# Fair and Optimal Decision Trees:
# A Dynamic Programming Approach

**Jacobus G.M. van der Linden**     **Mathijs M. de Weerdt**     **Emir Demirović**
Delft University of Technology, Department of Computer Science
`{J.G.M.vanderLinden, M.M.deWeerdt, E.Demirovic}@tudelft.nl`

## Abstract

Interpretable and fair machine learning models are required for many applications, such as credit assessment and in criminal justice. Decision trees offer this interpretability, especially when they are small. Optimal decision trees are of particular interest because they offer the best performance possible for a given size. However, state-of-the-art algorithms for fair and optimal decision trees have scalability issues, often requiring several hours to find such trees even for small datasets. Previous research has shown that dynamic programming (DP) performs well for optimizing decision trees because it can exploit the tree structure. However, adding a global fairness constraint to a DP approach is not straightforward, because the global constraint violates the condition that subproblems should be independent. We show how such a constraint can be incorporated by introducing upper and lower bounds on final fairness values for partial solutions of subproblems, which enables early comparison and pruning. Our results show that our model can find fair and optimal trees several orders of magnitude faster than previous methods, and now also for larger datasets that were previously beyond reach. Moreover, we show that with this substantial improvement our method can find the full Pareto front in the trade-off between accuracy and fairness.

## 1 Introduction

As machine learning (ML) is used in more domains that involve discrimination-sensitive decisions, demand for *fair*, *interpretable* and *accurate* models increases. ML, for example, is used in criminal justice [8], credit assessment [28], and housing appointments [7], each of which requires fair decisions. Simply removing the discrimination-sensitive attribute from the dataset does not necessarily result in less discrimination [33]. Instead, fairness is obtained by adding a constraint on the classifier: often either an individual or a group fairness constraint (see [10, 20] for a comparison of the two). The focus in this paper is on *group fairness*. With group fairness, a binary classifier is considered fair if the probability of receiving the positive outcome is the same for all protected groups. This fairness metric is formulated as a global constraint over the full dataset.

To increase confidence and prevent errors in these sensitive-decision domains, it also important that the ML models are interpretable [39]. Decision trees offer inherent interpretability and do not carry some of the risks of post-hoc attempts at explaining black-box models [36]. This is especially the case when decision trees are small [34]. On average, *optimal* decision trees, i.e., trees that for a given size provably maximize an objective such as accuracy, provide better out-of-sample performance than trees generated by heuristics when comparing trees of equal size [9, 15]. Therefore small optimal decision trees are ideal when searching for accurate and interpretable models.

There are several approaches for finding fair and optimal decision trees [2, 3, 25, 40], all of which use mixed integer programming (MIP). However, none of these models scale well. If fairness is not considered, small optimal decision trees can be found efficiently by using dynamic programming (DP)

methods that exploit the separability present in the tree structure [4, 5, 15, 23, 31]. Demirović and Stuckey [14] also use DP to find the Pareto front of optimal decision trees for biobjective nonlinear metrics.

However, a global fairness constraint cannot trivially be added to these algorithms, because the fairness constraint breaks the separability assumption by introducing dependency between subproblems. When optimizing for accuracy, for example, left and right subtrees can be solved independently after choosing what feature to branch on in the current decision node. In contrast, when optimizing for fairness, the left and right subtree cannot be optimized independently, since imbalance in one part of the tree can be cancelled out in another part of the tree.

Motivated by the success of DP methods for conventional optimal trees, we developed a novel DP method that successfully incorporates a global fairness constraint. We solve the subproblem dependency by computing upper and lower bounds on the final fairness values of partial solutions, thus enabling early pruning of the search space. Moreover, we present algorithmic improvements in the merging and comparing of subproblems, and perform a study of scalability. Overall our approach provides orders of magnitude improvements over state-of-the-art MIP methods.

This has several benefits. First, our algorithm can find fair and optimal decision trees for datasets that were previously too large to consider, often even within seconds. Second, it can find deeper and more accurate trees, while still satisfying the fairness constraint. Last, our algorithm can find the full Pareto-front in the trade-off between accuracy and fairness. This enables the responsible domain expert to make a well-informed decision and select a policy from the solution set that fits the context and application best.

As is commonly done in optimal decision tree search, we assume that all features have been binarized in advance [12]. Similarly, it is assumed that the label of every instance in the dataset is binary: either the positive/preferred outcome or the negative outcome. This is a common assumption in fair classification [16] and was also assumed in previous fair and optimal decision tree methods [25, 40].

The paper setup is as follows: we first discuss related work and preliminaries, and then we present our method. Finally, we evaluate the performance of the new method on twelve datasets from different application domains and we compare its performance to a leading MIP method from the literature.

## 2   Related work

This section discusses, in order, the related work of 1) algorithms for finding optimal decision trees, 2) MIP models for finding fair optimal decision trees, and finally 3) other (non-optimal) methods for finding fair decision trees.

**Algorithms for Optimal Decision Trees.**   The search for optimal decision trees is an NP-Hard problem [24]. Therefore decision tree search was traditionally done by means of heuristics such as the much-used CART algorithm [11]. Because of increasing computation power, searching for optimal decision trees has become feasible and Bertsimas and Dunn [9] showed that optimal decision trees can offer 1-5% better out-of-sample accuracy scores than previous heuristics. Their MIP model was consecutively improved by several others [41, 42].

During the same period, custom search and bound-based approaches, using techniques such as caching and branching, for finding (binary) classification trees were developed. These methods often outperformed the MIP models in terms of runtime and scalability, sometimes reducing runtimes of several hours to several seconds, by utilizing dynamic programming in a depth-first [4, 5, 15] or breadth-first approach [23, 31].

Of particular interest for this paper is the method presented in [14] which extends on these methods and finds decision trees for biobjective optimisation, minimizing the misclassification score for both classes in binary classification at the same time and returning the Pareto front of nondominated solutions. This method is further discussed in the preliminaries.

**Fairness in Optimal Decision Trees.**   Currently, the only methods for finding fair and optimal decision trees are MIP-based. Verwer and Zhang [40] developed a MIP model with a soft group fairness constraint in the objective. They report how their method can find a fair depth-three tree for a

dataset of 1000 instances in 15 minutes, when the model is given an initial solution from CART. But for the fair tree case they do not report being able to prove optimality within this time limit.

Aghaei et al. [2] present a MIP model that co-optimizes fairness and error for both classification and regression. Apart from group fairness, they also consider a notion of individual fairness separately. However, they are criticized by Ranzato et al. [35] that their model often provides accuracy not much better than a constant classifier. In [3] and [25], they continue their work and present a MIP model based on a max-flow formulation that results in a considerable improvement of the runtime. However, these models still have scalability issues because of the large number of binary variables.

**Other methods for Fair Decision Trees.** For completeness the following section mentions a number of related fair decision tree methods that do not guarantee a globally optimal solution. These can be divided into pre-, in-, and postprocessing methods.

Kamiran and Calders [26] present a preprocessing method that changes the labels in the dataset according to a group fairness metric. The updated dataset can then be used to train any regular classifier.

Some of the in-processing methods consider fairness in the classifier itself. Others use a regular classifier, combined with a meta- or master problem that takes care of the fairness constraint. Pedreschi et al. [33], for example, develop a method for finding fair association rules that can then be used to compile decision trees. Agarwal et al. [1] iteratively solve a number of cost-sensitive classification problems, with the costs determined by the number of violations of the fairness constraint. Grari et al. [21] propose a gradient tree boosting approach with an adversarial fairness constraint. Detassis et al. [17] use a MIP master problem that changes the labels for the next training iteration to satisfy the fairness constraint. There is, however, no guarantee of convergence. Valdivia et al. [38] developed an evolutionary algorithm that returns a Pareto front of decision trees with the multiobjective of both fairness and accuracy.

Kamiran et al. [27] test a CART-based approach with a split criterion that is a mix of information gain in the label and the sensitive feature. They did not observe a significant improvement when using this approach. As an alternative, they suggest a postprocessing relabeling method that uses a greedy knapsack algorithm to decide which leaf nodes to relabel to improve fairness at a minimum cost of accuracy.

**Summary of related work.** In summary, the current state-of-the-art has scalable custom methods for finding optimal conventional decision trees, but adding fairness constraints to these methods is non-trivial. MIP-based solutions exist for this problem, but they do not scale well. The alternative is to use heuristic methods, which result in suboptimal solutions.

## 3   Preliminaries

This section discusses the preliminaries of our method: an introduction of notation, a formal introduction of group fairness, a formal problem definition, and a dynamic programming formulation for optimal decision tree search that does not consider fairness.

**Notation.** A dataset $\mathcal{D}$ contains instances $(\mathbf{x}, a, y)$, with $\mathbf{x}$ the feature vector, $a$ the sensitive attribute and $y$ the label value. The number of instances in the dataset $\mathcal{D}$ is $N = |\mathcal{D}|$. Let $y_i$ be the label of instance $i$ and $a_i$ the value of the protected feature. Let $\mathcal{F}$ be the feature set and $f \in \mathcal{F}$ a feature in the feature set. We use $f_i$ to denote the value of this feature for instance $i$. Since we are only considering binary features and binary classification, we introduce the shorter notation $f_i$ and $\bar{f}_i$ to mean respectively that instance $i$ satisfies or does not satisfy feature $f$. The same idea applies for $y$ and $\bar{y}$, and $a$ and $\bar{a}$. We use subscript notation to refer to a subgroup of the dataset: for example, $\mathcal{D}_{\bar{f}}$ refers to all instances in dataset $\mathcal{D}$ that do not satisfy feature $f$; $N_a$ is the number of instances in the dataset that belong to group $a$, and $N_{y,\bar{a}}$ is the number of instances in the dataset with positive outcome that does not belong to group $a$.

**Group Fairness Formalized.** The specific group fairness definition considered in this paper is *demographic parity*, although our method can easily be extended to support other notions of group fairness, such as equality of opportunity. Demographic parity requires that the performance (the

expected percentage of positive outcomes) per group is the same [20]. More formally, if $\hat{y}$ is the predicted outcome, 1 is the positive outcome, and $a$ represents a (binary) sensitive/protected group, then demographic parity holds iff:

$$P(\hat{y} = 1 \mid a = 1) = P(\hat{y} = 1 \mid a = 0) \tag{1}$$

In binary classification, this constraint can be measured by considering the relative average performance of the two groups. Let $\hat{p}$ be the percentage of instances that receive the positive outcome, so $\hat{p}_a = N_{\hat{y},a}/N_a$. The difference in the performance between two groups is what we will call the imbalance $I = \hat{p}_a - \hat{p}_{\bar{a}}$. When the imbalance is limited to some value $\delta$, we say that it satisfies demographic parity up to bias $\delta$:

$$|I| = |\hat{p}_a - \hat{p}_{\bar{a}}| \le \delta \tag{2}$$

According to the definition of demographic parity, if a leaf node $n$ receives label 0, the partial imbalance $I_n$ in that node is also 0. Otherwise, the partial imbalance in a leaf node $n$ can be expressed as follows:

$$I_n = \frac{N_{n,a}}{N_a} - \frac{N_{n,\bar{a}}}{N_{\bar{a}}} \tag{3}$$

The partial imbalance of a branching node is the sum of the imbalance of both sub-nodes.

**Problem Definition.** The task of learning an optimal fair decision tree is to find the feature value tests in the branch nodes and the leaf node assignments for a tree of a given maximum size that minimize the number of misclassifications in a training data set while observing a fairness constraint. The percentage of correctly classified instances is called accuracy. This can be formalized as follows. The task is to find a decision tree classifier $h : \{0,1\}^{|\mathcal{F}|} \to \{0,1\}$ of depth $d$ that minimizes the misclassification score for a given dataset $\mathcal{D}$ while observing a fairness constraint up to bias $\delta$:

$$\min_h \sum_{(\mathbf{x},a,y)\in\mathcal{D}} |h(\mathbf{x}) - y| \tag{4}$$

$$\text{s.t.} \left| \sum_{(\mathbf{x},a,y)\in\mathcal{D}_a} \frac{h(\mathbf{x})}{N_a} - \sum_{(\mathbf{x},a,y)\in\mathcal{D}_{\bar{a}}} \frac{h(\mathbf{x})}{N_{\bar{a}}} \right| \le \delta$$

Alternatively, when searching for the full Pareto front, the parameter $\delta$ is not necessary, and the problem can be written as a multiobjective problem:

$$\min_h \left\{ \sum_{(\mathbf{x},a,y)\in\mathcal{D}} |h(\mathbf{x}) - y|, \left| \sum_{(\mathbf{x},a,y)\in\mathcal{D}_a} \frac{h(\mathbf{x})}{N_a} - \sum_{(\mathbf{x},a,y)\in\mathcal{D}_{\bar{a}}} \frac{h(\mathbf{x})}{N_{\bar{a}}} \right| \right\} \tag{5}$$

The Pareto front for Eq. 5 consists of all nondominated solutions $(M, I)$ with $M$ the misclassification score and $I$ the imbalance. A solution is considered dominated if there is another solution that performs better or similar on all objectives. Therefore, we can define the dominance relation $(\succ)$ for two solutions $(M_1, I_1)$ and $(M_2, I_2)$, and the function nondom as follows:

$$(M_1, I_1) \succ (M_2, I_2) \text{ iff } M_1 \le M_2 \wedge |I_1| \le |I_2| \wedge (M_1, I_1) \neq (M_2, I_2) \tag{6}$$

$$\text{nondom}(S) = \{s_1 \in S \mid \neg\exists\, s_2 \in S, s_2 \succ s_1\} \tag{7}$$

**Dynamic Programming Formulation for Decision Trees.** The problem of finding accurate decision trees has been formulated as a DP problem before in [15]. Of interest to this paper is the formulation for biobjective optimization in [14]. They minimize the misclassification score of two classes at the same time and return the Pareto front of nondominated solutions:

$$T_{\mathrm{BI}}(\mathcal{D}, d) = \begin{cases} \{(0, |\mathcal{D}_y|), (|\mathcal{D}_{\bar{y}}|, 0)\} & d = 0 \\ \text{nondom}\left(\cup_{f\in\mathcal{F}} \text{merge}\left(T_{\mathrm{BI}}(\mathcal{D}_f, d - 1), T_{\mathrm{BI}}(\mathcal{D}_{\bar{f}}, d - 1)\right)\right) & \text{else} \end{cases} \tag{8}$$

At the leaf node ($d = 0$), the function returns two solutions consisting of two values: the misclassification score per class (e.g., $(0, |D_y|)$ when label $\hat{y} = 0$ is selected, because all instances with $y = 0$ will be correctly classified and all instances with $y = 1$ will cause $|D_y|$ misclassifications for class 1). At a branching node ($d > 0$), the function should return the merge of two solution sets and retain only the nondominated solutions resulting from that merge. The function merge combines two sets of partial solutions (from the left and right subtree) by element-wise addition:

$$\text{merge}(S_1, S_2) = \{(a_1 + a_2, b_1 + b_2) \mid (a_1, b_1) \in S_1, (a_2, b_2) \in S_2\} \tag{9}$$

# 4 Method Description

In this section, we present our method DPF (DP Fair) and show how to deal with a global fairness constraint that introduces subproblem dependency. Similar to Eq. 8, we define a recursive function $T_F$ that returns a set of all nondominated (partial) solutions, resulting in a full Pareto front of fair and optimal decision trees. In order to guarantee that the method returns the full Pareto front we must do an exhaustive search of all combinations of subtrees, pruning only those solutions for which we can prove that they will never be part of an optimal (nondominated) solution.

**Dependency.** The recursive function 8 introduced in the preliminaries assumes that the left and the right tree can be independently optimized, but this is not the case when considering group fairness, which is a global constraint on the whole tree. When comparing two possible subtrees, one of which discriminates against group A and the other discriminating against group B, it is not clear which one is better, because it will depend on what happens in the rest of the tree. Even when both subtrees discriminate against the same group A, it is not immediately clear that the one with lower discrimination score will be better, but again it will depend on what happens in the rest of the tree.

More formally, the previous proposed biobjective method requires objectives to be *additive* and *monotonic* in order to create independent subproblem. The additive property holds if two partial solutions $(a, b)$ and $(a', b')$ can be combined by addition: $(a + a', b + b')$. The monotonic property holds for a biobjective function $f(a, b)$ if for any two possible different inputs $a, b$ and $a', b'$, the following dominance relation holds:

$$a \leq a' \wedge b \leq b' \rightarrow f(a, b) \succ f(a', b') \tag{10}$$

However, because fairness is measured by the absolute value of the imbalance, both the additive property and the monotonic property are not satisfied. Our method addresses this problem.

**Upper and lower bounds for fairness.** The key idea in our method is to compute upper and lower bounds for the imbalance value of partial solutions to enable comparison. When upper and lower bounds $\bar{I}_R$ and $\underline{I}_R$ are known for the imbalance in the rest ($R$) of the tree, then the final imbalance value of a tree that contains node $n$ will be in the range: $[\underline{I}_R + I_n, \bar{I}_R + I_n]$. The lower bound ($\underline{I}$) and the upper bound ($\bar{I}$) for the final *absolute* imbalance value can now be computed:

$$\underline{I}(I_n, [\underline{I}_R, \bar{I}_R]) = \begin{cases} 0 & \text{if } 0 \in [\underline{I}_R + I_n, \bar{I}_R + I_n] \\ \min\left(|\underline{I}_R + I_n|, |\bar{I}_R + I_n|\right) & \text{else} \end{cases} \tag{11}$$

$$\bar{I}(I_n, [\underline{I}_R, \bar{I}_R]) = \max\left(|\underline{I}_R + I_n|, |\bar{I}_R + I_n|\right) \tag{12}$$

In Eq. 11, the lower bound is zero if the permitted range contains zero; otherwise it is the minimum absolute value in that range. In Eq. 12, the upper bound is the maximum absolute value in the permitted range.

With these upper and lower bounds on the final absolute imbalance value, we can redefine the dominance relation of Eq. 6:

$$\begin{aligned} (M_1, I_1) \succ (M_2, I_2)) \text{ iff } & (M_1, I_1) \neq (M_2, I_2) \\ & \wedge M_1 \leq M_2 \\ & \wedge \left(\bar{I}(I_1, [\underline{I}_R, \bar{I}_R]) \leq \underline{I}(I_2, [\underline{I}_R, \bar{I}_R]) \vee I_1 = I_2\right) \end{aligned} \tag{13}$$

A solution is dominated by another solution if 1) the other solution has a lower misclassification score; and 2) either has an upper bound on the absolute imbalance lower than the lower bound on the absolute imbalance of this solution, or has precisely the same imbalance score.

**Finding the Pareto front of optimal fair trees.** Now we can define a new function for finding fair trees $T_F(\mathcal{D}, d, [\underline{I}_R, \bar{I}_R])$ that optimizes Eq. 5:

$$T_F(\mathcal{D}, d, [\underline{I}_R, \bar{I}_R]) = \begin{cases} \text{leaf}(\mathcal{D}, [\underline{I}_R, \bar{I}_R]) & d = 0 \\ \text{branch}(\mathcal{D}, d, [\underline{I}_R, \bar{I}_R]) & \text{else} \end{cases} \tag{14}$$

The functions leaf and branch will be introduced next. Both use the nondom function, which is now based on our new dominance comparison as described in Eq. 13 (but for ease of notation, $[\underline{I}_R, \bar{I}_R]$ will often be left out).

In a leaf node $n$, the function should return the following $(M, I)$ solutions, with imbalance values based on Eq. 3:

$$\text{leaf}(\mathcal{D}, [\underline{I}_R, \bar{I}_R]) = \text{nondom}\left(\left\{(|\mathcal{D}_y|, 0), \left(|\mathcal{D}_{\bar{y}}|, \frac{N_{n,a}}{N_a} - \frac{N_{n,\bar{a}}}{N_{\bar{a}}}\right)\right\}\right) \tag{15}$$

Eq. 15 returns up to two solutions, each described by two values: first the misclassification score as before, and second the imbalance when assigning the positive outcome.

In a branching node, the data set is split on some feature $f$, resulting in two subproblems. Each subproblem is solved independently by passing the best known bounds for the other subproblem. Let $U(\mathcal{D}_{\bar{f}}, [\underline{I}_R, \bar{I}_R], d-1)$ denote the function that returns these best known bounds. These bounds can be based on dataset inspection, or based on previous (cached) solutions. Bounds from inspecting the dataset can be derived by assigning one label to all instances of one group and the other label to all other instances and vice versa. The results provide the maximum discrimination that could possibly happen. When (cached) partial solutions for the rest of the tree are known, the bounds can be derived by taking the minimum and maximum imbalance values among solutions. With these bounds, the solutions from the two subproblems are then merged as follows:

$$\begin{aligned}
\text{branch}(\mathcal{D}, d, [\underline{I}_R, \bar{I}_R]) = \text{nondom}\,(\cup_{f \in \mathcal{F}}\,\text{merge}\,( \\
T_F(\mathcal{D}_{\bar{f}}, d-1, \quad U(\mathcal{D}_f, [\underline{I}_R, \bar{I}_R], d-1)), \\
T_F(\mathcal{D}_f, d-1, \quad U(\mathcal{D}_{\bar{f}}, [\underline{I}_R, \bar{I}_R], d-1))))
\end{aligned} \tag{16}$$

**Finding a single best tree.** It is also possible with this same DP formulation to search for a single optimal and fair tree up to bias $\delta$, as defined in Eq. 4. This can be achieved by pruning all (partial) solutions and retaining only those that are *possibly* fair. Then, in the root node of the search, the tree with minimum misclassification score is selected from the list of solutions. A solution is possibly fair if the lower bound on the absolute imbalance value is less than or equal to $\delta$. For this, we introduce the function prune, which should filter a set of solutions $S$ before it is passed to the nondom function.

$$\text{prune}(S, [\underline{I}_R, \bar{I}_R]) = \left\{(M_n, I_n) \mid (M_n, I_n) \in S,\ \underline{I}(I_n, [\underline{I}_R, \bar{I}_R]) \leq \delta\right\} \tag{17}$$

**Pseudocode.** This section gives the pseudocode for DPF. In addition to the description above, the pseudocode also considers upper and lower bounds and cache. Therefore, the function $T_F$ receives the current best known upper bound ub on the misclassification score as an extra parameter.

The function prune must be redefined to take into account this upper bound on the misclassification score.

$$\begin{aligned}
\text{prune}(S, \text{ub}, [\underline{I}_R, \bar{I}_R]) = \{(M_n, I_n) \mid (M_n, I_n) \in S, \\
\underline{I}(I_n, [\underline{I}_R, \bar{I}_R]) \leq \delta, M_n < \text{ub}\}
\end{aligned} \tag{18}$$

For shorter notation, define:

$$\text{filter}(S, \text{ub}, [\underline{I}_R, \bar{I}_R]) = \text{nondom}\left(\text{prune}\left(S, \text{ub}, [\underline{I}_R, \bar{I}_R]\right), [\underline{I}_R, \bar{I}_R]\right) \tag{19}$$

Also we define the functions $\underline{M}$ to return the best misclassification score within a set of solutions:

$$\underline{M}(S) = \min\{M \mid (M, I) \in S\} \tag{20}$$

Furthermore, let $LB$ return the best known lower bound on the misclassification score for a subproblem, or zero if no such lower bound is known.

With these changes, see Algorithm 1 for the resulting pseudocode of DPF. In this algorithm $S$ is the set of solutions, and lb and ub are the lower and upper bounds, with subscripts $L$ and $R$ denoting left and right subtrees. For a full Pareto front, the algorithm must be called with a fairness cut-off value of $\delta = 1$.

**Other algorithmic improvements.** There are a number of other improvements in DPF for reducing the runtime. To reduce the runtime cost of the merge function, before merging, the two sets of partial solutions are first pruned based on bounds on the misclassification score and discrimination score derived from the other set. Then the two sets are sorted by misclassification score to enable early termination of the merge if it can be proven that all consecutive combinations of partial solutions will

**Algorithm 1:** Tree search of depth $d$ with fairness on a dataset $\mathcal{D}$ for a feature set $\mathcal{F}$.

---

$T_\text{F}(\mathcal{D}, d, \text{ub}, [\underline{I}_R, \bar{I}_R])$

    **if** $d = 0$ **then**

        $S \leftarrow \left\{ (|\mathcal{D}_y|, 0), \left( |\mathcal{D}_{\bar{y}}|, \frac{N_{n,a}}{N_a} - \frac{N_{n,\bar{a}}}{N_{\bar{a}}} \right) \right\}$

        **return** $\text{filter}(S, \text{ub}, [\underline{I}_R, \bar{I}_R])$

    $\langle S, \text{lb}, \text{stat} \rangle \leftarrow \text{cache}[\mathcal{D}, d, [\underline{I}_R, \bar{I}_R]]$

    **if** $\text{lb} \geq \text{ub}$ **then return** $\emptyset$

    **if** $\text{stat} = \text{optimal}$ **then return** $\text{filter}(S, \text{ub}, [\underline{I}_R, \bar{I}_R])$

    $S \leftarrow \emptyset$

    **for** $f \in \mathcal{F}$ **do**

        $\text{lb}_R \leftarrow \text{LB}(\mathcal{D}_f, d-1, [\underline{I}_R, \bar{I}_R])$

        $S_L \leftarrow T_\text{F}(\mathcal{D}_{\bar{f}}, d-1, \text{ub} - \text{lb}_R, U(\mathcal{D}_f, [\underline{I}_R, \bar{I}_R], d-1))$

        **if** $S_L = \emptyset$ **then continue**

        $\text{lb}_L \leftarrow \text{LB}(\mathcal{D}_{\bar{f}}, d-1, [\underline{I}_R, \bar{I}_R])$

        $S_R \leftarrow T_\text{F}(\mathcal{D}_f, d-1, \text{ub} - \text{lb}_L, U(\mathcal{D}_f, [\underline{I}_R, \bar{I}_R], d-1))$

        **if** $S_R = \emptyset$ **then continue**

        $S \leftarrow S \cup \text{prune}\left( \text{merge}(S_L, S_R, [\underline{I}_R, \bar{I}_R]), \text{ub}, [\underline{I}_R, \bar{I}_R] \right)$

    $S \leftarrow \text{nondom}(S)$

    **if** $S = \emptyset$ **then**

        $\text{cache}[\mathcal{D}, d, [\underline{I}_R, \bar{I}_R]] \leftarrow \langle \emptyset, \text{ub}, \text{lower bound} \rangle$

        **return** $\emptyset$

    $\text{cache}[\mathcal{D}, d, [\underline{I}_R, \bar{I}_R]] \leftarrow \langle S, \underline{M}(S), \text{optimal} \rangle$

    **return** $S$

---

exceed the misclassification upper bound. In the root node of the search, the solutions are sorted on the imbalance value to enable faster elimination of non-fair solutions.

The runtime of the $\text{nondom}$ function can be reduced by observing that partial solutions with a lower bound imbalance of $0$ do not need to be compared with other solutions because they will always be nondominated. For the sake of brevity, all these have been left out of the method description. See our code repository for full implementation details.[1]

Similar to [14, 15] our method also uses a special depth-two solver to increase the runtime performance. Our source code contains the full details. Appendix A presents a complexity analysis of DPF. Appendix B shows how our algorithm can easily be modified to also optimize tree sparsity.

## 5 Experimental Results

The following section analyzes the performance of DPF. We focus on the analysis of the runtime and scalability of DPF: 1) How does DPF compare to the state-of-the-art in terms of runtime? 2) What impact do our pruning method and merge improvements have on the runtime? 3) What is the runtime cost to find the full Pareto front? 4) What impact do parameters such as number of features, instances, etc. have on the runtime of DPF?

For an out-of-sample analysis, see Appendix D.

**Experiment setup.** We evaluate DPF on all datasets mentioned in the survey by Le Quy et al. [30]. The data preprocessing and binarization is also done as described in that paper. Categorical variables are encoded through one-hot encoding, except for variables with twenty or more categories, which were removed (this applies to the KDD census dataset). The binarized datasets are included in our code repository (if the license permits redistribution). The Diabetes and Law-school dataset are left out of the evaluation because the best tree of depth three is (almost) identical to the trivial solution (assign the majority label to all instances). See Appendix C for more details.

---

[1] https://gitlab.tudelft.nl/jgmvanderlinde/dpf

The algorithm receives the whole dataset as input and is asked to find a tree with an unfairness limit of $\delta = 1\%$. In a later experiment, we also examine the impact of this parameter on the runtime. The values shown are the average of five runs and a time limit of one hour is set for every run. All experiments are run on a 2.6Ghz Intel i7 CPU with 8GB RAM using only one thread.

**The FairOCT model**  To our knowledge, the only methods available for finding fair and optimal decision trees are MIP models [2, 25, 40]. Verwer and Zhang [40] limit their model to solving problems with less than 1000 instances. In our initial comparison of [2] and [25], the latter outperformed the first by a large margin. Therefore, we compare our method with their FairOCT model. They model the decision tree as a flow graph where all instances must flow from the source (connected to the root node of the tree) to one of the sink nodes (one for each class), to which all nodes are connected. In our experiments, the FairOCT model is solved with Gurobi 9.0 using the default parameters.

**Runtime comparison.**  Table 1 shows the runtime comparison between FairOCT and our method DPF. DPF can find optimal trees for $d = 2$ within one second for all datasets, about four to five orders of magnitude faster than FairOCT. FairOCT is able to find $d = 2$ trees only for the smallest datasets within one hour. It struggles with an increasing number of data instances because every data instance is represented by a binary variable. Also for $d = 3$, DPF can find optimal trees within one second for several datasets and all problems are solved within a minute. For $d = 4$, the exponential nature of the problem becomes clearly visible. Datasets with both large number of features and instances are no longer solvable within one hour. However, DPF still succeeds in finding the best tree for a number of datasets, among which Adult, with over 45000 instances.

**The impact of our novel pruning method and the merge improvements**  Table 2 shows the runtime results when DPF is run 1) without our novel pruning method (not applying the prune and nondom methods of Eqs. 7 and 17; 2) without our improvements to the merge method; 3) without our improvement to the nondom method, and 4) to generate a full Pareto front.

For several datasets, our pruning method decreases the runtime by one or more orders of magnitude. This is specifically the case for depth four, where the subproblem dependency is more significant than for depth three. Our method can find optimal trees for depth four for eight out of twelve datasets within one hour, but without the pruning method, this is possible for only four datasets.

Our improvements to the merge function in some cases significantly reduce the runtime. This is specifically the case for datasets that have a high initial bias (the bias of an optimal tree if fairness would not be considered): Adult, COMPAS recid. and Dutch census have an initial bias of 18%, 16%, and 14% respectively, and for these datasets the merge improvements reduce the runtime by an order of magnitude. The other data sets in Table 2 have an initial bias of 1-6% and the merge improvements result in only a small reduction of the runtime. The improvements to the nondom function are also essential for the performance of our pruning mechanism.

Table 1: Runtime comparison of DPF and FairOCT. Runtime is in seconds. FairOCT resulted in an out of memory error for KDD census income, so no runtime can be reported.

| | | | | FairOCT | DPF | | |
|---|---|---|---|---|---|---|---|
| Dataset | $a$ | $|\mathcal{D}|$ | $|\mathcal{F}|$ | $d = 2$ | $d = 2$ | $d = 3$ | $d = 4$ |
| Adult | Gender | 45222 | 17 | > 1h | < 1 | < 1 | 1012 |
| Bank Marketing | Married | 45211 | 46 | > 1h | < 1 | 2 | > 1h |
| Communities & Crime | Race | 1994 | 97 | > 1h | < 1 | 4 | 1261 |
| COMPAS recid. | Race | 6172 | 9 | > 1h | < 1 | < 1 | 9 |
| COMPAS viol. recid. | Race | 4020 | 9 | 1594 | < 1 | < 1 | < 1 |
| Dutch census | Gender | 60420 | 58 | > 1h | < 1 | 5 | > 1h |
| German credit | Gender | 1000 | 69 | > 1h | < 1 | 2 | 741 |
| KDD census income | Race | 284556 | 117 | - | < 1 | 32 | > 1h |
| OULAD | Gender | 21562 | 45 | > 1h | < 1 | 3 | > 1h |
| Ricci | Race | 118 | 4 | < 1 | < 1 | < 1 | < 1 |
| Student-Mathematics | Gender | 395 | 55 | 284 | < 1 | < 1 | 24 |
| Student-Portuguese | Gender | 649 | 55 | 866 | < 1 | < 1 | 32 |

Table 2: DPF runtimes without the pruning and dominance checks (¬P), without the merge improvements (¬M), without the nondom improvements (¬D), for finding the full Pareto-front (Pr), and for comparison the default runtime (Df). Datasets for which DPF has a default runtime below one second or over one hour, are left out.

| | $d = 3$ | | | | | | $d = 4$ | | | | |
|---|---|---|---|---|---|---|---|---|---|---|---|
| Dataset | ¬P | ¬M | ¬D | Pr | Df | Dataset | ¬P | ¬M | ¬D | Pr | Df |
| Bank | 5 | 3 | 319 | 4 | 2 | Adult | > 1h | > 1h | > 1h | > 1h | 1012 |
| Com.&Cr. | 13 | 5 | 19 | 7 | 4 | Com.&Cr. | > 1h | 1329 | 2473 | 1939 | 1261 |
| Dutch | > 1h | 101 | 1814 | 510 | 5 | COMP. r. | > 1h | 394 | 1658 | 1659 | 9 |
| German | 8 | 2 | 22 | 2 | 2 | German | > 1h | 770 | 1971 | 738 | 741 |
| KDD | 144 | 35 | 1254 | 50 | 32 | Stud. Math | 144 | 24 | 29 | 412 | 24 |
| OULAD | 5 | 4 | 1448 | 3 | 3 | Stud. Port. | 1834 | 32 | 38 | 259 | 32 |

**Generating the full Pareto-front**   Because of the runtime improvements of DPF, it is also able to find a *full*, fair and optimal decision tree Pareto front. In contrast, Valdivia et al. [38] find a partial and non-optimal Pareto front and the FairOCT model is used to find an optimal, but still partial, Pareto front. Table 2 shows the runtime performance of DPF when tasked to find the full Pareto front. In comparison to FairOCT, DPF is several orders of magnitude faster. For COMPAS recid., for example, DPF found the Pareto front for depth two consisting of 29 solutions in $0.12 \pm 0.04$ seconds. The FairOCT method generates the partial Pareto front by repeatedly solving the problem with a different maximum bias, resulting in a runtime several times higher than the values reported in Table 1.

See Figure 1 for a number of full Pareto fronts generated by DPF for depth 2-4. These plots confirm the trade-off between accuracy and fairness as reported in previous works.

**Method evaluation.**   To provide more insight into our method we evaluate the impact of the number of features, the number of dataset instances, the minimum leaf node size and the maximum allowed bias in the fairness constraint on the runtime of DPF for three datasets when searching for trees of depth three. Figure 2 shows the results. It can be observed from these experiments that the number of features is the most important factor, resulting in an exponential increase, which is in line with previous studies [15, 31]. One might expect that the method would scale linearly with increasing dataset size, but the experiments show almost no increase in runtime after a certain size. The reason for this is that the runtime of the method is mostly dependent on the number of unique partial solutions that need to be merged by Eq. 9. This number of unique solutions is strongly dependent on the number of features and is only weakly related to the number of instances.

DPF can easily be changed to limit the minimum leaf node size, by returning an empty set in Eq. 15 when the number of instances does not exceed the required leaf node size. Increasing the minimum leaf node size decreases the amount of unique partial solutions, and consequently we see a significant decrease in the runtime when the minimum leaf node size is increased. Adding such a minimum leaf node size often does not have a significant impact on the accuracy score.

Relaxing the fairness constraint to values larger than $\delta = 1\%$ causes the runtime to increase for the Dutch census dataset. This can be explained because a more relaxed constraint means a larger search space and less strong fairness bounds. However, when the constraint is even more relaxed, the problem

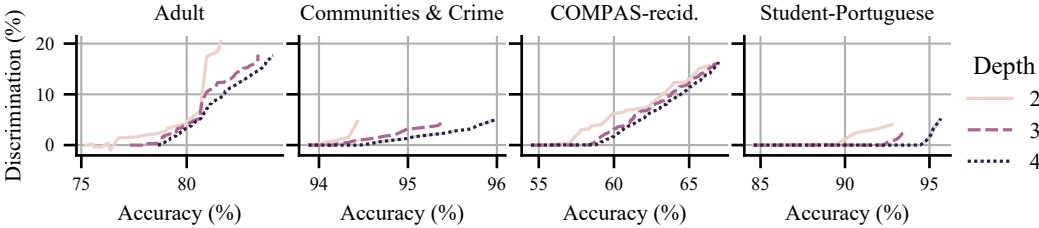

Figure 1: The full Pareto front of training accuracy and discrimination for four datasets for trees of depth 2, 3 and 4, generated by DPF.

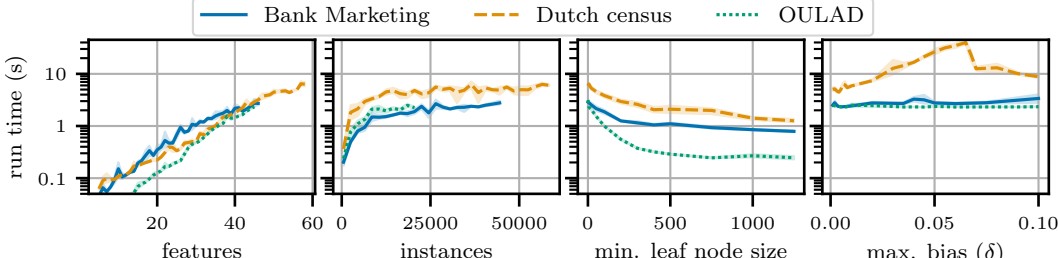

Figure 2: The impact of the number of features, instances, minimum leaf node size and maximum allowed bias in the fairness constraint, $\delta$, on the runtime of DPF with $d = 3$. The shaded area shows the values within one standard deviation. Note the logarithmic scale of the y-axis.

is closer to a normal optimal tree search, and the accuracy bounds become stronger, again resulting in a shorter runtime. For the Bank marketing and OULAD datasets, the default discrimination when no fairness constraint is in place, is lower: $1.2\%$ and $3.0\%$, versus Dutch census: $14.1\%$. This explains why these datasets show almost no difference in runtime when changing the value $\delta$.

## 6 Conclusion

We show how dynamic programming can be used to find fair and optimal decision trees, even when considering a global fairness constraint that introduces subproblem dependency. We solve the subproblem dependency by computing upper and lower bounds on the final fairness value and thus enable comparison of partial solutions. The results show that our method DPF can find fair optimal trees several orders of magnitude faster than the state-of-the-art. As a result, DPF succeeds in finding fair and optimal decision trees even for large datasets that were previously beyond reach. Moreover, we present the first specialized algorithm for finding fair optimal decision trees that can also find the full Pareto front in terms of accuracy and fairness.

An extension of DPF would be to support other group-based notions of fairness, such as equality of opportunity. To better deal with unbalanced datasets, another extension of the presented research would be to replace the accuracy objective with an objective that can cope better with unbalanced datasets, for example, with the biobjective method presented in [14], or by using balanced accuracy. Future work could also investigate the possibility of combining top-down search with bottom-up search. Finally, based on the evaluation results, new heuristic methods could be investigated, for example, by considering not all possibly-fair partial solutions, but only a representative subset.

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
