# Supplementary Material for
# Fair and Optimal Decision Trees:
# A Dynamic Programming Approach

**Jacobus G.M. van der Linden**     **Mathijs M. de Weerdt**     **Emir Demirović**
Delft University of Technology, Department of Computer Science
`{J.G.M.vanderLinden, M.M.deWeerdt, E.Demirovic}@tudelft.nl`

## A   Complexity analysis

The runtime of DPF as presented in Algorithm 1 depends on many factors and specifically on the number of feasible solutions and the amount of solutions that can be pruned early in the search. We here provide a worst case bound that assumes that all trees are feasible and no solution can be pruned. An upper bound of the number of unique complete trees of depth $d$ is given by $n_t = O(|\mathcal{F}|^{2^d-1}2^{2^d})$: i.e., the product of the number of possible branching decision assignments and leaf node label assignments. In the worst case scenario, no tree would be pruned and the nondom operation would compare every tree with every other tree: $n_t^2$ comparisons. A second term in the runtime complexity is from the recursive tree search calls. For $d > 0$, DPF has $2|\mathcal{F}|$ recursive calls, resulting in a total of $2^d|\mathcal{F}|^d$ calls to DPF. Each of these calls traverses the dataset once. Therefore a worst case runtime complexity for DPF is: $O(n_t^2 + 2^d|\mathcal{F}|^d|\mathcal{D}|)$. In practice, however, the runtime is much smaller because of pruning.

## B   Non-complete trees

For the sake of brevity, the main text of this paper only shows how to search for complete trees, that is, trees of depth $d$ with $2^d - 1$ branch nodes and $2^d$ leaf nodes. Our method also allows to search for smaller trees, as was done similarly in [3].

The DP formulation in Eq. 16 can be extended as follows to also allow for incomplete/sparse trees. Here $n$ signifies the number of nodes in the tree.

$$
\begin{aligned}
\text{nondom}\,\big(\text{prune}\,\big(&\cup_{f\in\mathcal{F},i\in[0,n-1]}\text{merge}\,\big(\\
&T_F(\mathcal{D}_{\bar{f}}, d-1, n-i-1),\quad U(\mathcal{D}_f, [\underline{I}_R, \bar{I}_R], d-1, i)),\\
&T_F(\mathcal{D}_f, d-1, i),\quad U(\mathcal{D}_{\bar{f}}, [\underline{I}_R, \bar{I}_R], d-1, n-i-1))\big)\big)\big)
\end{aligned}
\tag{1}
$$

With this change in place, the solver can search for incomplete trees. This also allows to add a parameter $\alpha$ to prevent overfitting. The misclassification score now becomes $M + \alpha n$. The parameter $\alpha$ describes how much the misclassification score should at least decrease in order to justify adding another node to the tree. The addition of this parameter to the algorithm is trivial.

## C   Dataset details

Table 1 shows detailed information about every dataset considered in this study. The references for the original datasets can be found here [1, 2, 4, 5, 8, 10, 11].

36th Conference on Neural Information Processing Systems (NeurIPS 2022).

Table 1: Datasets used for evaluation. Preprocessed as described in [9]. Training accuracy (%) and discrimination results (%) are shown for the best tree of depth three that does not consider fairness when training with the full dataset. The sign of the discrimination score tells which of the two groups is discriminated against.

| Name | $|\mathcal{D}|$ | $|\mathcal{F}|$ | protected feature | y=1 $a=1$ | y=1 $a=0$ | y=0 $a=1$ | y=0 $a=0$ | Acc. d=3 | Disc. d=3 |
|------|------|------|------|------|------|------|------|------|------|
| Adult | 45222 | 17 | Gender | 9539 | 1669 | 20988 | 13026 | 83.4 | -17.8 |
| Bank | 45211 | 46 | Married | 2755 | 2534 | 24459 | 15463 | 90.0 | 1.2 |
| Com.&Cr. | 1994 | 97 | Race | 1017 | 855 | 7 | 115 | 95.4 | -4.5 |
| COMP. r. | 6172 | 9 | Race | 1281 | 2082 | 822 | 1987 | 66.7 | -16.4 |
| COMP. v.r. | 4020 | 9 | Race | 1285 | 2083 | 174 | 478 | 84.1 | -1.6 |
| Dutch | 60420 | 58 | Gender | 18860 | 9903 | 11287 | 20370 | 81.4 | -14.1 |
| German | 1000 | 69 | Gender | 499 | 201 | 191 | 109 | 75.3 | -5.5 |
| KDD | 284556 | 117 | Race | 15926 | 1475 | 223155 | 44000 | 94.4 | -1.1 |
| OULAD | 21562 | 45 | Gender | 7727 | 6928 | 3841 | 3066 | 69.1 | -3.0 |
| Ricci | 118 | 4 | Race | 41 | 15 | 27 | 35 | 100 | -30.3 |
| Stud. Math | 395 | 55 | Gender | 132 | 133 | 55 | 75 | 93.4 | -6.3 |
| Stud. Port. | 649 | 55 | Gender | 216 | 333 | 50 | 50 | 93.7 | 3.4 |

# D  Test Evaluation

**Experiment setup.** The evaluation in the main text is focused on analyzing the runtime of our DPF method. This section further analyzes the out-of-sample performance of DPF and compares it with two heuristics. The pre-processing (massaging) approach presented in [6], and a post-processing approaches proposed in [7]. We will call the pre-relabelling method KamPre and the post-relabelling method KamPost, after their first author (Kamiran).

KamPre pre-processes the training data by relabeling a number of instances in the training data such that the training data no longer is biased. Like them, we use a Naive Bayes classifier to decide which labels to change. Unlike them, we use the newly labeled data to train a standard decision tree with CART.

KamPost differs from CART by using a different splitting criterion and by its post-relabelling of the leaf nodes. KamPost uses a splitting criterion that is a mix of information gain in the class label and information gain in the sensitive attribute (IGC+IGS). After generating the tree, it heuristically changes the label of some leaves in such a way that discrimination is minimized with the least loss of accuracy.

We do not compare to the optimal MIP method FairOCT because its optimal solutions would either be the same as those generated by DPF, or -if multiple optimal solutions exists- any difference could only be attributed to a random selection of one out of several optimal models.

We first evaluate the out-of-sample performance by running DPF for $d = 2, 3, 4$ and KamPost for $d = 3, 4$ for different maximum allowed bias $\delta = 1\%, 5\%, 100\%$ on three datasets that are commonly evaluated in the literature. Note that KamPost with $\delta = 100\%$ is the same as plain CART. We run each case 10 times on random stratified train-test splits of $75\%$ vs $25\%$. We here do not compare to KamPre because it does not take a maximum allowed bias as an input factor. Figure 1 shows the resulting distribution of test accuracy and test discrimination. The sign of the discrimination score tells which of the two groups is discriminated against.

**Evaluation.** From the results in Figure 1 it can be observed that the optimal decision trees generated by DPF in general have better out-of-sample accuracy than KamPost for the same or smaller depth. In several cases DPF $d = 2$ even outperforms KamPost, even though it uses four times less nodes. However, in general the variance in test accuracy is high.

The variance in test discrimination is also high, and both methods often exceed the imposed discrimination limit in the test evaluation. This problem is less visible with the Adult dataset, probably because of its larger number of data instances.

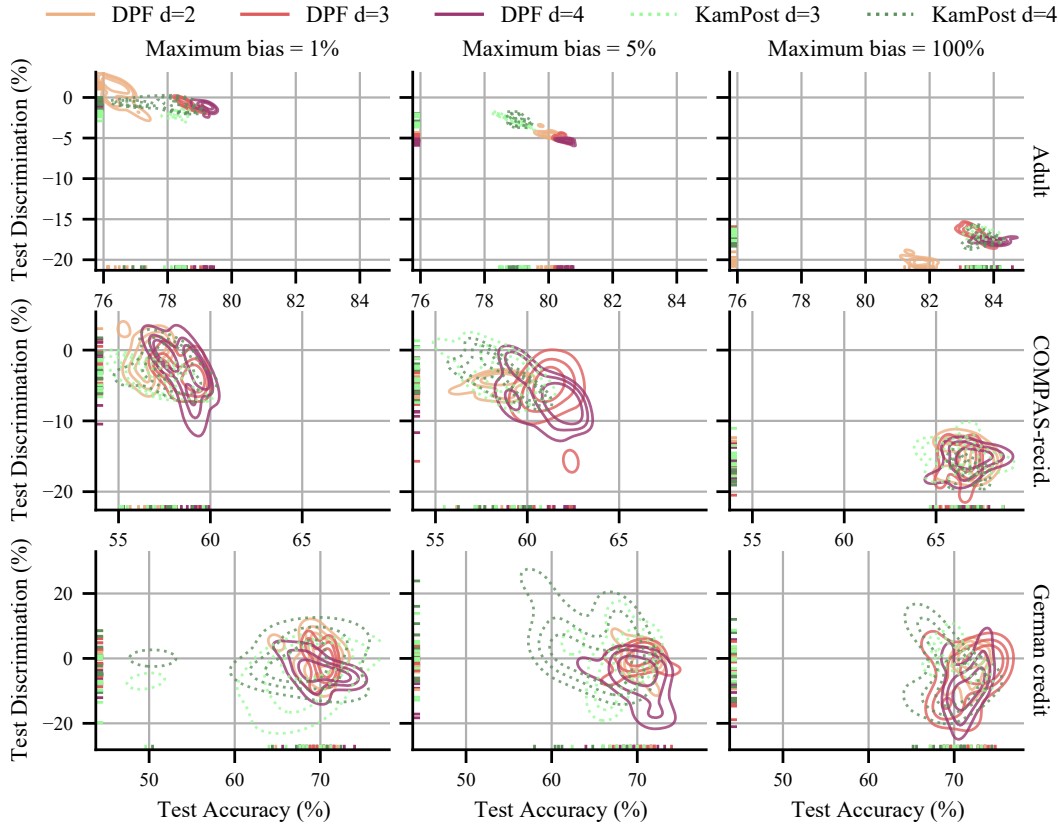

Figure 1: Out-of-sample accuracy and discrimination for three datasets, for maximum bias $\delta = 1\%, 5\%, 100\%$. The figure shows the distribution of 10 runs.

For both the COMPAS-recid. and German credit dataset almost maximum accuracy can already be obtained with a decision tree of depth two, so the addition of extra nodes does not help much. For the Adult dataset, however, deeper trees can provide better accuracy.

Table 2: Out-of-sample average accuracy and discrimination $\pm$ the standard deviation (%) for solutions with a maximum training bias of $\delta = 5\%$ and a maximum depth of $d = 3$. Whenever the 5% discrimination threshold is exceeded on average, the result is marked in red. Best performing accuracy score per dataset is marked bold, if significantly better than other methods that also stay within the 5% discrimination limit (p-value $< 5\%$).

| | DPF | | KamPre | | KamPost | |
|---|---|---|---|---|---|---|
| Dataset | Accuracy | Disc. | Accuracy | Disc. | Accuracy | Disc. |
| Adult | $80.4 \pm 0.3$ | $-5.1 \pm 0.4$ | $75.2 \pm 0.0$ | $0.0 \pm 0.0$ | $\mathbf{78.0} \pm 1.5$ | $-1.9 \pm 1.1$ |
| Bank | $\mathbf{89.7} \pm 0.1$ | $1.7 \pm 0.6$ | $89.1 \pm 0.3$ | $0.9 \pm 0.6$ | $89.0 \pm 0.4$ | $0.5 \pm 0.3$ |
| Com.&Cr. | $\mathbf{94.6} \pm 0.3$ | $-3.2 \pm 1.0$ | $93.0 \pm 1.1$ | $-3.3 \pm 2.6$ | $93.9 \pm 0.3$ | $-1.3 \pm 2.3$ |
| COMP. r. | $\mathbf{59.1} \pm 2.4$ | $-4.6 \pm 3.1$ | $61.4 \pm 1.8$ | $-9.8 \pm 2.8$ | $55.6 \pm 1.9$ | $-1.1 \pm 2.1$ |
| COMP. v.r. | $83.7 \pm 0.2$ | $-0.3 \pm 0.5$ | $82.2 \pm 1.1$ | $-4.0 \pm 2.1$ | $83.8 \pm 0.1$ | $-0.3 \pm 0.8$ |
| Dutch | $\mathbf{77.4} \pm 0.3$ | $-5.0 \pm 1.0$ | $76.0 \pm 0.2$ | $-9.9 \pm 0.4$ | $68.2 \pm 10.9$ | $-0.6 \pm 0.6$ |
| German | $70.3 \pm 1.7$ | $-2.1 \pm 3.0$ | $69.8 \pm 1.3$ | $-0.7 \pm 3.9$ | $70.3 \pm 1.0$ | $-0.9 \pm 3.4$ |
| KDD | $\mathbf{94.3} \pm 0.0$ | $-1.0 \pm 0.1$ | $93.9 \pm 0.0$ | $0.0 \pm 0.0$ | $93.9 \pm 0.0$ | $0.0 \pm 0.1$ |
| OULAD | $\mathbf{68.7} \pm 0.3$ | $-2.1 \pm 1.0$ | $68.2 \pm 0.6$ | $-1.8 \pm 1.1$ | $68.0 \pm 0.1$ | $0.1 \pm 0.2$ |
| Ricci | $66.0 \pm 4.4$ | $-13.4 \pm 12.3$ | $100.0 \pm 0.0$ | $-28.1 \pm 0.0$ | $53.3 \pm 0.0$ | $0.0 \pm 0.0$ |
| Stud. Math | $\mathbf{85.5} \pm 6.6$ | $-2.8 \pm 8.3$ | $89.7 \pm 2.8$ | $-5.5 \pm 3.2$ | $73.9 \pm 11.8$ | $-2.5 \pm 4.2$ |
| Stud. Port. | $90.5 \pm 2.8$ | $0.8 \pm 5.0$ | $91.2 \pm 4.2$ | $6.1 \pm 4.9$ | $89.4 \pm 4.2$ | $2.3 \pm 3.1$ |

**Tuned for number of nodes.** In our next analysis, we prevent overfitting during training for all three methods by using 10 random validation splits of 25% of the training data to find what number of branching nodes is best. The tree size with the best average accuracy in the validation set, while on average respecting the discrimination constraint is selected as best. The full training dataset is then used to generate a tree of that size. Table 2 shows the results when all three algorithms are used to find trees of a maximum bias of 5%.

**Discussion.** DPF searches for optimal decision trees, which means it will always find the tree with maximum accuracy for the training dataset, and thus always outperform heuristics on performance in the training dataset. The results in Table 2 show that when DPF is tuned for selecting the right number of nodes, this on average also generalizes to better performance than KamPre and KamPost in the test set.

We compare the results of the methods that achieve (on average) a test discrimination lower than 5%, and among those select the method with highest test accuracy. There are seven datasets for which DPF is significantly better than KamPre and KamPost ($p < 5\%$), with differences in accuracy even as large as 11.6% (Student-Mathemtacis) or 9.2% (Dutch census). KamPost only scores best for Ricci and Adult. Ricci is the smallest of all datasets with only 118 instances and 4 features, but KamPost's result is only slightly better than random. KamPost also performs best for Adult, with DPF exceeding the limit by 0.1%. For three datasets, no method is significantly better than the others.

The results also confirm the findings from Figure 1 that the variance in the discrimination value is often high, specifically for the small datasets. This means that for those instances it is difficult to generalize and overfitting in terms of discrimination is still happening. It is an open question how this can be reduced.