# OpenReview forum: "Fair and Optimal Decision Trees: A Dynamic Programming Approach"
_NeurIPS.cc/2022/Conference — NeurIPS 2022 Accept_

### Official Review · Reviewer_Vs5A · 2022-07-07

**Rating:** 5
**Confidence:** 3
**Soundness:** 3 good
**Presentation:** 3 good
**Contribution:** 2 fair

**Summary:**

The paper address the problem of finding fair and accurate decision trees. Using demographic parity as the fairness definition, they try to minimize misclassification while at the same time satisfying a fairness constraint. The problem is hence formulated as a multiobjective problem in which both misclassification error and group imbalance are minimized. Having defined a dominating solution as one which is similar or better in both objectives, they search for Pareto fronts of nondominated solutions.

Previous work have proposed dynamic programming approaches for biobjective optimization in decision trees, and finding the Pareto front  of nondominated solutions in the case where the objectives are additive and monotonic. Since the imbalance part of their objective is not monotonic (contains absolute value), the dynamic programming approach cannot be directly used. As a result, the authors propose a way to calculate the upper and lower bound of the imbalance value, and later on, redefine the dominance relation based on these values. They then propose a dp approach that finds the Pareto frontiers and improve the time complexity of the search by pruning the search space based on the upper and lower bound values of the imbalance.

**Questions:**

I would like to thank the authors for taking the time to write the paper in a well-organized way. It was my pleasure to review the paper. I have a few questions from the authors:

- How good are the set of Pareto frontiers? Can it be that a solution that is not dominated by any other solution is not really a good one? Assume the accuracy is very high but it is at the cost of very low fairness (in the tuple (M,I), M is very high and I is very low. Then it could happen that no other solution can dominate this one because no solution can beat M, but the solution is very imbalanced.

- In equation 11 and 12, is it not the case that the lower bound is always equal to the lowerbound of the rest + I_n and the upper bound is equal to the upperbound of the rest + I_n? I do not understand why you wrote it with min/max?

- it is not very easy to understand equations 14-17. What is the function U exactly? How exactly does the merge function work? if it works the same as defined in Eq.9, then how is it using the upper/lower bound information? If it is not using that information, why is it input to the function?


**Limitations:**

they have provided the run time of their approach so it gets clear that it gets higher as the dataset or the number of features grows. Maybe they could also talk about space complexity and how much storage they used to store the intermediate solutions, since they use a dp approach. They could also show the performance considering more fairness criteria rather than just demographic parity. Their approach is applicable to binary classification with binary features. It would be nice to mention what happens if the features are not binary and whether the algorithm could scale.

**Strengths And Weaknesses:**

Strengths
- very well written except for some typos mentioned below
- they consider a well-motivated problem

Weaknesses

- From section 4 on, the paper does not flow very well. In the end of section 3, it gets clear that due to non-monotonicity of the imbalance function, the dp approach cannot be directly used and the reader is waiting for that problem to be addresses. In the beginning of section 4 you directly mention that you present DPF, without saying what it stands for, and without saying how it tries to overcome the monotonicity problem. Section 4 is hard to follow. Not everything is clearly defined such as the meaning of (R), being the rest of the tree, and how exactly you calculate the upper/lower bounds. I believe the quality of the upper/lower bounds directly affect the quality of the solutions found so I do not find the sentence "These bounds can be trivial, ..." in line 202 really convincing.

There are further presentation issues mentioned below:
- punctuations before/after equations are completely missing
- typo in the first line of Eq.8
- Eq 8 needs some explanation. You copy it from another paper without introducing what it is searching for. For example, it is not clear what (|D|,0) is showing. I had to refer to the cited papers to understand but I think it is better if the paper is self-contained and the reader does not need to search for the meaning of notation elsewhere.
- lines 176-178 are not well written
- line 191, "is" is missing
- you never mention what is "DPF". You use the abbreviation from the first occurrence.

---

> ### Author Response · Authors · 2022-07-30
> **Author response**
>
> Dear reviewer,
>
> Thank you for your review of our work. Thank you also for your detailed feedback on some of our writing. Based on your comments we have been able to improve the clarity of our writing and explain better how this work contributes to the literature. We will here provide some detailed responses. (Your comments are put in italics, and our response follows afterward.)
>
>
> 1. _From section 4 on, the paper does not flow very well. (...) I believe the quality of the upper/lower bounds directly affect the quality of the solutions found so I do not find the sentence "These bounds can be trivial, ..." in line 202 really convincing._
>
> Thank you for the feedback on our method section. We have critically reexamined it and changed the order of explanation based on your feedback. We have also included a clearer explanation of how the bounds are obtained.
>
> See also our response to your question 3 (Q3) under point 4 below.
>
>
> 2. _(Q1) How good are the set of Pareto frontiers? Can it be that a solution that is not dominated by any other solution is not really a good one?_
>
> The reviewer is correct that deciding which Pareto optimal solution to select in a generic way is difficult. We believe the responsible domain-expert is the only person that can decide on the trade-off between accuracy and fairness, since the specific application and context need to be considered. Therefore, the generation of the Pareto front is precisely one of the advantages of our method because it assists the domain-expert in better decision making. We mentioned this in the introduction, but will make it more clear.
>
>
> 3. _(Q2) In equation 11 and 12, is it not the case that the lower bound is always equal to the lowerbound of the rest + I_n and the upper bound is equal to the upperbound of the rest + I_n? I do not understand why you wrote it with min/max?_
>
> The min/max is necessary because we are calculating the lower and upper bounds for the absolute value of fairness. Consider a lower bound of the rest -8 and an upper bound +2, and a fairness value of the current node +1, then the final fairness value will be in the range [-7,+3]. The minimum absolute value in this range is 0 (according to Eq. 11), and its maximum is 7 (according to Eq. 12).
>
> To make this more clear we explained these equations more clearly in the text.
>
>
> 4. _(Q3) It is not very easy to understand equations 14-17. What is the function U exactly? How exactly does the merge function work? if it works the same as defined in Eq.9, then how is it using the upper/lower bound information? If it is not using that information, why is it input to the function?_
>
> The function U returns the best known bounds for a subtree. This function returns either:
> * The best known bounds based on cached solutions (e.g., when the other subtree has already been examined).
> * Bounds based on dataset inspection. E.g., what would the worst fairness value be if all instances from one group would receive a positive label, and all instances from the other group would receive a negative label (or the other way around).
>
> The merge function indeed does not use the upper/lower bound information. In Eq. 15 is passed to $T_F$, not to the merge function.
>
> Based on your feedback we have made this more clear in the method section.
>
>
>
> 5. _Maybe they could also talk about space complexity and how much storage they used to store the intermediate solutions, since they use a dp approach._
>
> A worst case bound for the space complexity is given by the number of possible distinct trees, since we store all subproblem solutions in the cache. This number of distinct trees is precisely expressed by Hu et al. [23]. A simple loose upper bound is $O(f^{2^d-1})$.
>
> In our experiments, however, we have never reached space limitations.
>
> We will add a discussion of runtime and space complexity in the appendix.
>
>
>
> 6. _They could also show the performance considering more fairness criteria rather than just demographic parity._
>
> Our contribution indeed enables the consideration of other global non-monotonic constraints, when upper and lower bounds can be expressed. This means that our method is applicable to a broader spectrum of constraints. We mention the extension to other notions of fairness in our future work. Specifically, other group-based notions of fairness, such as equality of opportunity, can be considered by the method we propose.
>
>
> 7. _Their approach is applicable to binary classification with binary features. It would be nice to mention what happens if the features are not binary and whether the algorithm could scale._
>
> Our approach would also work for multi-valued decision trees, but the branching factor would increase. In our work, we assume that non-binary features are binarized first, which is a common assumption in other optimal fair works, e.g., [25].
>
> Finally we want to thank the reviewer again for reading and reviewing our paper and for the helpful comments.
>
> The authors

---

### Official Review · Reviewer_Bp3p · 2022-07-09

**Rating:** 7
**Confidence:** 4
**Soundness:** 4 excellent
**Presentation:** 4 excellent
**Contribution:** 3 good

**Summary:**

This paper proposes a dynamic programming (DP) model to construct optimal decision trees subject to fairness constraints, i.e., where the positive classification observes demographic parity. The methodology extends an existing bi-objective recursive reformulation for decision trees and returns the full Pareto frontier in terms of the optimal and fairness criteria. In particular, the authors develop upper and lower bounds on the fairness constraint to ensure all generated solutions are feasible, in addition to pruning techniques to speed up the frontier enumeration. The numerical study evaluates the approach against current (mostly MIP-based) state-of-the-art methods.

**Questions:**

1. Given the nice performance, is it possible to extend the approach to non-binary trees? It would not be surprising to find cases where the DP would perform well (if the state space is still relatively "compact").

2. Could authors discuss the scalability of the approach for datasets beyond the ones presented?

3. Labelling approaches in multiobjective DP optimization can be significantly improved when using bidirectional search, such as the one presented in [1]. That is, the state-space graph of the DP is constructed from top-down and bottom-up separately, and the Pareto frontier is "merged" when the two layers meet. This is the key performance in state-of-the-art multiobjective models, such as the ones appearing in shortest-path subproblems for vehicle routing. I wonder how this relates to Algorithm 1, or whether that could help improve the methodology?

[1] Galand L, Ismaili A, Perny P, Spanjaard O (2013) Bidirectional preference-based search for state space graph problems. Proceedings of the Sixth International Symposium on Combinatorial Search

**Strengths And Weaknesses:**

Strengths
- Very strong numerical results, significantly outperforming the state of the art
- Idea is intuitive and easy to implement
- Paper very well written

Weaknesses
- I have concerns about the contribution; except for the lower or upper bounds, the methodology itself seems to be derived in a somewhat straightforward way from existing works
- Scalability and usage could be detailed more thoroughly

Major comments

Overall, I greatly enjoyed reading the work because the methodology is intuitive, involves interesting bounding procedures associated with the structure of the fairness constraint, and the numerical results are quite strong, outperforming MIP-based models very prominently. However, I have two major concerns after reading the work.

(i) [Novelty.] My (possibly wrong) impression is that the authors essentially implemented the ideas from two existing works (references [14] and [15] of the paper) for this paper. More precisely, they specialized the bi-objective approach to accommodate fairness constraints by considering interval-based labelling to ensure the full Pareto frontier was enumerated correctly. While the effectiveness is apparent and there is some novelty in the upper/lower bounds, this brought me some concerns about whether the results are somewhat incremental to existing literature. I believe this could be addressed in multiple ways. For instance, authors could expand on other constraint classes that could leverage the methodology, or whether there are other fairness definitions that could also be accommodated by the technique.

(ii) [Scalability and Usage.] It is impressive that the authors are enumerating the full Pareto frontier. This shines light into two usual questions that also follow any type of multiobjective work. First, it is unclear how scalable that is; while it worked well for the datasets in the numerical experiments, how large are the problems that actually can be solved by DP in this context? Second, even if the Pareto frontier can be fully enumerated, what would be the guidelines/insights to pick the appropriate tree among such a possibly large set? It would be great if authors could expand further on those concepts.

---

> ### Author Response · Authors · 2022-07-30
> **Author response**
>
> Dear reviewer,
>
> Thank you for your review of our work. Based on your comments we have been able to improve the clarity of our writing and explain better how this work contributes to the literature. We will here provide some detailed responses.
>
>
> 1. _(i.a) [Novelty.] (...)My (possibly wrong) impression is that the authors essentially implemented the ideas from two existing works (references [14] and [15] of the paper) for this paper._
>
> The non-trivial novelty in our work is that our method can optimize a non-monotonic global constraint. In contrast, previous work [14,15] requires monotonic objectives. Further note that the bounds are only one of our contributions, we also introduced several algorithmic techniques to provide notable speed ups (Table 2), and a study of scalability (Figure 2). Overall our approach provides orders of magnitude improvements over previous approaches.
>
> This is currently explained in the introduction and conclusion, and we will explain more clearly in our method section why solving subproblems with a shared constraint is not trivial.
>
>
> 2. _(i.b) [The] authors could expand on other constraint classes that could leverage the methodology, or whether there are other fairness definitions that could also be accommodated by the technique._
>
> Our contribution indeed enables the consideration of other global non-monotonic constraints, when upper and lower bounds can be expressed. This means that our method is applicable to a broader spectrum of constraints. We mention the extension to other notions of fairness in our future work. Specifically, other group-based notions of fairness, such as equality of opportunity, can be considered by the method we propose.
>
>
> 3. _(ii.a) [Scalability and Usage.] It is impressive that the authors are enumerating the full Pareto frontier (...) how large are the problems that actually can be solved by DP in this context?_
>
> Thank you for this comment on the impressiveness of enumerating the whole Pareto frontier. We respond to your questions as part of our response to your question 2 (Q2) below in point 6.
>
>
> 4. _(ii.b) (...) what would be the guidelines/insights to pick the appropriate tree among such a possibly large set? _
>
> We believe the responsible domain-expert is the only person that can decide on the trade-off between accuracy and fairness, since the specific application and context need to be considered. Our method now makes it possible to enumerate the Pareto front to assist in better decision making. We mentioned this in the introduction, but will make it more clear.
>
>
> 5. _(Q1) Given the nice performance, is it possible to extend the approach to non-binary trees?_
>
> Our approach would also work for multi-valued decision trees, but the branching factor would increase. In our work, we assume that non-binary features are binarized first, which is a common assumption in other optimal fair works, e.g., [25]. Our method could also be extended to multi-labeled decision trees. However, when considering fairness, a binary label is what is typically considered (the preferred outcome vs the non-preferred outcome).
>
>
> 6. _(Q2) Could authors discuss the scalability of the approach for datasets beyond the ones presented?_
>
> The total number of possible distinct trees is the most important factor in its scalability (as also stated in [15]). The worst case runtime complexity of DPF is $O(T n )$, with $n = |D|$ the size of the dataset, and $T$ the number of possible distinct trees. This number is precisely expressed by Hu et al. [23]. A simple upper bound is $T \leq O(f^{2^d-1})$. Therefore we can state that $O(nf^{2^d-1})$ is a loose bound on the runtime.
>
> In our experimental runtime analysis in Figure 2 we confirm these findings: dataset size is not very important, and by reducing the number of possible distinct trees, we observe a drop in runtime.
>
> We will add a discussion of runtime complexity in the appendix, thank you for the suggestion.
>
>
> 7. _(Q3) Labelling approaches in multiobjective DP optimization can be significantly improved when using bidirectional search, such as the one presented in [1]. (...) I wonder how this relates to Algorithm 1, or whether that could help improve the methodology?_
>
> This is an interesting idea. Thank you for suggesting it. It seems that the intuition behind the algorithm presented in [1] is that the search is symmetrical, i.e., source and target nodes could be swapped and the problem would be the same. This, however, is not the case with optimal decision tree search: root and leaf nodes are not symmetrical, and the number of possible leaf nodes is exponential.
>
> Overall, including this idea would require further research into how a bottom-up search in decision tree search would work. We will consider this as future work; thank you for the suggestion.
>
> Finally we want to thank the reviewer again for reading and reviewing our paper and for the helpful comments.
>
> The authors

---

> > ### Comment · Reviewer_Bp3p · 2022-08-09
> > **Response feedback**
> >
> > Thank you for the detailed and thorough response, you have clarified my doubts.

---

### Official Review · Reviewer_5CeZ · 2022-07-09

**Rating:** 5
**Confidence:** 1
**Soundness:** 3 good
**Presentation:** 2 fair
**Contribution:** 3 good

**Summary:**

The authors designed a specialized algorithm, DPF(Dynamic Programming Fair?) that can find fair optimal decision trees, i.e., the decision trees with the best performance metrics under a given fairness gap constraint. The algorithm can also return a Pareto front of performance and fairness and use the front to filter out dominated splits at each level. Dynamic programming is used in DPF to find possible splits. The authors show numerical results that DPF is much more efficient to train compared to related works like FairOCT.

**Questions:**

1. In some of the other decision tree models, putting samples into different bins help significantly with the split finding and training speed. In this work, when keeping all the non-dominated submodels, is it possible to use this technique?
2. If the technique in question 1 is possible, what are some of the conditions that guarantee when splitting at the bin boundaries results in two non-dominated submodels, all other splits within this bin are also non-dominated? Is this somehow related to the monotonic condition?
3. Is it possible to come up with theoretical reasonanings on the efficiency of DPF? Can you specify when, or on what type of dataset DPF can perform much better than the related works and when not?
4. DPF is compared to other in-process fairness oriented algorithms in the article, so a natural question to ask is can DPF be compared to or combined with pre-process fairness oriented methods or post-process fairness oriented methods?

**Limitations:**

As discussed in the weaknesses already.

**Strengths And Weaknesses:**

Strengths:
1. Strong numerical results showing that DPF is much more efficient to train compared to related works like FairOCT
2. Clear Problem statement

Weaknesses:
1. No theorem or propositions to show the advantage of DPF theoretically
2. Pseudo code would be nice to have in the main article than in the appendix to help the readers understand the algorithm

---

> ### Author Response · Authors · 2022-07-30
> **Author response**
>
> Dear reviewer,
>
> Thank you for your review of our work. Based on your comments we have been able to improve the clarity of our writing and explain better how this work contributes to the literature. We will here provide some detailed responses. (Your comments are put in italics, and our response follows afterwards.)
>
>
>
> 1. _No theorem or propositions to show the advantage of DPF theoretically._
>
> (As we have also answered the review by Reviewer NysV, point 2, Q1)
>
> Theoretical guarantees of optimal trees on out-of-sample accuracy and fairness is an important open question, both in our work and other related work. However empirically it has been shown that optimal trees do provide better results on out-of-sample accuracy [9,14,15]. This is also the case when considering fairness as both [25] and our results (Table 3) show.
>
> See also our response to your question 3 (Q3) in point 4 below.
>
>
>
> 2. _Pseudo code would be nice to have in the main article than in the appendix to help the readers understand the algorithm_
>
> We agree with the reviewer that moving the pseudo-code from appendix to the main paper would be an improvement. We will do so in the camera-ready version with the extra page.
>
>
>
> 3. _(Q1) In some of the other decision tree models, putting samples into different bins help significantly with the split finding and training speed. In this work, when keeping all the non-dominated submodels, is it possible to use this technique?_
>
>     _(Q2) If the technique in question 1 is possible, what are some of the conditions that guarantee when splitting at the bin boundaries results in two non-dominated submodels, all other splits within this bin are also non-dominated? Is this somehow related to the monotonic condition?_
>
>
> In our approach we assume the binarization is performed as a preprocessing step, and we adopt the binarization as done in [30], as also explained in our experimental setup. For this, or any other binarization, our model will provide the optimal Pareto front.
>
> Furthermore, we would like to highlight that our method scales linearly (in the worst case) for the number of samples, reducing the need for putting samples into bins.
>
>
>
> 4. _(Q3) Is it possible to come up with theoretical reasonanings on the efficiency of DPF? Can you specify when, or on what type of dataset DPF can perform much better than the related works and when not?_
>
> Intuitively the advantage of our method is that we exploit the structure of decision trees which eliminates symmetries, and enables the reuse of computed subproblems, both due to our dynamic programming formulation, as explained in our introduction section. These points are difficult to include in MIP formulations. Moreover, our method is not much impacted by the size of the dataset (Figure 2), whereas in MIP formulations new binary variables need to be introduced for each dataset instance.
>
> Note that based on our experiments, our method is orders-of-magnitude faster than the competing methods, and we have not found a dataset where other approaches would outperform our method.
>
>
>
> 5. _(Q4) DPF is compared to other in-process fairness oriented algorithms in the article, so a natural question to ask is can DPF be compared to or combined with pre-process fairness oriented methods or post-process fairness oriented methods?_
>
> The focus of our work is on computing optimal trees faster than previous optimal (in-processing) methods, and therefore we highlight the improvements in scalability in our experimental results section.
>
> In the appendix we compare our method DPF to the in/post-processing method of Kamiran [25] and show the advantage of DPF.
>
> In response to the reviewer’s question, yes, we have now also compared DPF to the preprocessing method presented in Kamiran and Calders in 2009 [26], which “massages” the training data by changing the labels such as to remove bias from the training data. These results have now been included in Table 3 in the appendix and also clearly show the advantage of DPF over this method.
>
> Finally we want to thank the reviewer again for reading and reviewing our paper and for the helpful comments.
>
> The authors

---

### Official Review · Reviewer_NysV · 2022-07-11

**Rating:** 4
**Confidence:** 4
**Soundness:** 3 good
**Presentation:** 3 good
**Contribution:** 2 fair

**Summary:**

The authors studied the optimal and fair decision trees. They proposed a dynamic programming method considering the fairness constraint. Due to the global fairness constraint, the decision tree is not separable. Using upper and lower bounds on the fairness values, the authors decrease the search space and improve the solution time.

**Questions:**

Is there a theoretical guarantee that obtaining the "global" optimal solution to the posed problem would lead to higher accuracy and/or fairness for the out-of-sample instances?

What is the impact of highly imbalanced datasets on the performance of the proposed algorithm?

The authors have reported only the runtimes for DPF and FairOCT. How about the accuracies and fairness scores? In case of multiple optimal solutions, one would think that the out-of-sample performances may differ. Do these methods obtain exactly same solutions?

How do the computation times of the proposed method compare against the method of Kamiran? Are the results for Kamiran in Table 3 obtained after tuning the hyperparameters of the method?

The following recent paper discusses fairness in optimal trees by formulating MIP. Their results seem promising. It would be great if the authors also review this paper and highlight their differences:

Jo, N., Aghaei, S., Benson, J., Gómez, A., & Vayanos, P. (2022). Learning Optimal Fair Classification Trees. arXiv preprint arXiv:2201.09932.



**Limitations:**

Limitations are discussed in the conclusion section.

**Strengths And Weaknesses:**

With the use of upper and lower bounds on fairness, the authors set forth a new dominance relation. The same bounds also lead to a new pruning mechanism. Consequently, the authors obtain a method that finds the Pareto front, and it is faster than the other methods proposed in the literature.

The paper is mainly based on two papers [14, 15] that authors have also cited. Adding only group fairness constraint and pruning the trees through trivial bounds are straightforward extensions. In its current form, the work does not introduce enough novelty to the literature.

---

> ### Author Response · Authors · 2022-07-30
> **Author response**
>
> Dear reviewer,
>
> Thank you for your review of our work. Based on your comments we have been able to improve the clarity of our writing and explain better how this work contributes to the literature. We will here provide some detailed responses. (Your comments are put in italics, and our response follows afterwards.)
>
>
>
> 1. _Adding only group fairness constraint and pruning the trees through trivial bounds are straightforward extensions._
>
> The non-trivial novelty is that our method can optimize a non-monotonic global constraint. In contrast, previous work [14,15] requires monotonic objectives. Further note that the bounds are only one of our contributions, we also introduced several algorithmic techniques to provide notable speed ups (Table 2), and a study of scalability (Figure 2). Overall our approach provides orders of magnitude improvements over previous approaches.
>
> This is currently mentioned in the introduction and conclusion, and we will explain more clearly in our method section why solving subproblems with a shared constraint is not trivial: it is no longer possible to determine which solution(s) are best, without considering the rest of the tree.
>
>
>
> 2. _(Q1) Is there a theoretical guarantee that obtaining the "global" optimal solution to the posed problem would lead to higher accuracy and/or fairness for the out-of-sample instances?_
>
> Theoretical guarantees of optimal trees on out-of-sample accuracy and fairness is an important open question, both in our work and other related work. However empirically it has been shown that optimal trees do provide better results on out-of-sample accuracy [9,14,15]. This is also the case when considering fairness as both existing work [25] and our results (Table 3) show.
>
>
>
> 3. _(Q2) What is the impact of highly imbalanced datasets on the performance of the proposed algorithm?_
>
> Our current experiments contain at least two datasets that are highly imbalanced (majority class with more than 90% of the instances: KDD census income and Communities & Crime). We will make this more clear by including in the appendix a table that describes per dataset the number of positive and negative instances.
>
> Intuitively, imbalanced datasets are easier since the initial bounds on misclassification score can be more tight.
>
>
>
> 4. _(Q3) The authors have reported only the runtimes for DPF and FairOCT. How about the accuracies and fairness scores? In case of multiple optimal solutions, one would think that the out-of-sample performances may differ. Do these methods obtain exactly same solutions?_
>
> Our focus is on computing optimal fair trees faster. Therefore runtime is the main metric for comparison since both our method and FairOCT solve exactly the same problem to optimality.
>
> In Appendix C Table 3 and Figure 3 we report on the out-of-sample accuracy and fairness of DPF and a heuristic. Based on another reviewer's comment we have added another heuristic. We did not include a comparison with FairOCT because DPF and FairOCT obtain the same optimal solution (or “randomly” select a solution from the same set of optimal solutions). We have made this more clear in Appendix C.
>
>
>
> 5. _(Q4) How do the computation times of the proposed method compare against the method of Kamiran? Are the results for Kamiran in Table 3 obtained after tuning the hyperparameters of the method?_
>
> The method of Kamiran et al. has negligible runtime, but note that it does not provide optimality guarantees. The Kamiran method was hypertuned in the same way as the DPF method (tuned for the number of nodes), as is mentioned in the text in appendix C.
>
>
>
> 6. _The following recent paper discusses fairness in optimal trees by formulating MIP. Their results seem promising. It would be great if the authors also review this paper and highlight their differences: Jo, N., Aghaei, S., Benson, J., Gómez, A., & Vayanos, P. (2022). Learning Optimal Fair Classification Trees. arXiv preprint arXiv:2201.09932._
>
>
> The reviewer pointed out a recent promising paper. We discuss the referenced paper in our work, see FairOCT [25]. Our results show orders-of-magnitude improvements over the referenced paper (see Table 1).
>
> Finally we want to thank the reviewer again for reading and reviewing our paper and for the helpful comments.
>
> The authors

---

### Meta-Review · Area_Chair_4G7N · 2022-08-23

**Recommendation:** Accept
**Confidence:** Certain

**Metareview:**

I recommend acceptance due to the strengths identified by the positive reviews, despite some doubts expressed by more negative reviews. This paper modifies existing dynamic programming approaches for learning decision trees to accommodate non-monotonic constraints, motivated in particular by group fairness. Experiment show that this approach is orders of magnitude faster than existing alternatives.

The main unresolved reviewer concern is novelty---how much of a contribution is the ability to hand non-monotonic constraints? If this paper were exclusively targetted to the decision tree community, this would be an important concern. However, I view the significance of this contribution in terms of making decision trees a computationally tractable option for designing fair decisions. Interpretability is an important concern in many domains where fairness is an issue, and thus this is an important contribution.

**Award:**

No

---

### Decision · Program_Chairs · 2022-09-14

Accept